# Loss of polarity by Cdc42 depletion and oncogenic Kras activation in the mouse intestinal epithelia leads to a necrotizing enterocolitis (NEC)-like disease

Zheng Zhang[1], Cuiqing Fan [1], Ryan Jorgensen [1], Pamela Sylvestre[2], Mike Adam [1] & Yi Zheng [1] ✉

Cell polarity is essential for maintaining intestinal epithelial organization and function. Here we show that combined loss of polarity by epithelial loss of Cdc42 with oncogenic Kras expression in mice causes small intestine failure leading to weight loss, inflammation, epithelial necroptosis, and lethality. These phenotypic defects are characterized by a loss of intestinal stem cells, disrupted epithelial architecture, altered hippo signaling, elevated inflammatory cytokines, and activation of necroptotic cell death, that closely resemble necrotizing enterocolitis (NEC). Single-cell transcriptomic analysis reveals a coordinated dysregulation of polarity machinery, inflammatory pathways, and necroptosis program. Suppression of YAP, IL-1, TNFα signaling or necroptosis rescues the intestinal pathology. Similar NEC-like phenotypes arise when Cdc42 loss and oncogenic Kras activation are initiated from intestinal stem cells. These findings provide mechanism insights involving polarity-YAP-IL1/TNFα signaling induced necroptosis for the synergistic effect of hyperactivation of Kras signaling and loss of polarity in disrupting intestinal epithelia.

Cdc42 is a member of the Rho GTPase family widely recognized for its role in regulating actin cytoskeleton dynamics, adhesion/cell junction signaling, and polarity[1–3]. In the intestine, Cdc42 is particularly important for maintaining epithelia polarity, which is essential for the formation and function of the crypt-villus axis, as loss of Cdc42 in intestinal epithelial cells leads to a broad tissue hyperplasia accompanied by hyperproliferation of transient amplifying (TA) cells and intestinal stem cells (ISCs), crypt enlargement, and defective Paneth cell differentiation and localization in the intestine[4]. Recent studies further highlighted a role for Cdc42 in regulating the transition of intestinal stem cells to transit-amplifying cells, implicating it in intestinal homeostasis and ISC differentiation[5]. Interestingly, despite dramatic hyperproliferation of the Cdc42-null crypt cells due to

upregulated hippo signaling, intestinal epithelia Cdc42 knockout (KO) mice do not develop tumors and have a mostly normal life expectancy[4].

Oncogenic KRAS mutations are commonly found in colorectal cancer (CRC), resulting in constitutive activation of the Ras signaling pathway, and are causal for uncontrolled cell proliferation and survival[6–8]. The KRAS G12D mutation has been shown to drive the development of CRC by activating downstream signaling pathways, including RAF/MEK/ERK, to promote unchecked cell growth and resistance to apoptosis[9,10]. While the KRAS mutation is pivotal in the progression and maintenance of CRC, they act in conjunction with additional genetic and epigenetic alterations, and the progression to CRC requires further genetic mutations[11,12].

[1]Division of Experimental Hematology and Cancer Biology, Cancer and Blood Diseases Institute, Children's Hospital Medical Center, University of Cincinnati College of Medicine, Cincinnati, OH, USA. [2]Division of Pathology, Children's Hospital Medical Center, University of Cincinnati College of Medicine, Cincinnati, OH, USA. ✉e-mail: yi.zheng@cchmc.org

Since loss of polarity is a hallmark of intestinal cancer, we are interested in investigating the combined effect of Cdc42 loss and oncogenic Kras G12D in the intestinal epithelia, as either mutation can induce hyperplasia, and the hyperproliferation induced by Kras G12D and loss of polarity caused by Cdc42 KO might synergize to drive a transformation to carcinoma. To better understand how the combined loss of polarity, a known phenotype in epithelial cancers, with oncogenic Kras mutation, we have generated a genetic mouse model with simultaneous inducible Cdc42 deletion and oncogenic Kras G12D activation. Surprisingly, these mice developed necrotizing enterocolitis (NEC)-like defects, which is one of the most common and devastating gastrointestinal conditions primarily affecting premature infants, characterized by inflammation and necrosis of the bowel[13,14], leading to lethality of the mice within a short time frame. Corroborating this observation, an analysis of human colon adenocarcinoma datasets from the Cancer Genome Atlas (TCGA) database reveals that the expression of KRAS mutants and defects in polarity-related molecules, including Cdc42, Dlg5, and Scrib, are mutually exclusive in colon cancer patients. We further identify elevated YAP signaling and IL-1/TNFα mediated inflammation and necroptosis in the mouse model as key mediators of the development of NEC-like defects. Our study highlights an interaction between Cdc42 loss and oncogenic Kras G12D in the intestinal epithelium, which, instead of promoting proliferation and transformation, results in a severe NEC-like condition.

## Results

### Cdc42 deletion combined with oncogenic Kras G12D mutation in intestinal epithelium disrupts epithelial morphology and causes lethality

To generate a mouse model to study the combined effects of Cdc42 KO and oncogenic Kras mutation in the intestine, intestinal epithelium-specific inducible villin-CreERT2 (villin-CreER) expressing mice were crossbred with Cdc42 flox/flox mice[4], and with the Kras$^{LSL-G12D}$ mice to generate villin-CreER; Cdc42$^{flox/+}$(hereafter referred to as WT) mice, villin-CreER; Cdc42$^{flox/flox}$ (hereafter referred to as Cdc42 KO) mice, villin-CreER; Kras$^{LSL-G12D}$ (hereafter referred to as Kras) mice, and villin-CreER; Cdc42$^{flox/flox}$; Kras$^{LSL-G12D +}$ (hereafter referred to as Cdc42 KO/Kras) mice. The Cdc42 KO/Kras mice became severely ill after tamoxifen (TAM) injections to run on the mutations. We analyzed the morphology of the Cdc42 KO/Kras mice and found a severely disrupted intestinal epithelium in both the small intestine and colon that showed broken villi and tissue hypoplasia (Fig. 1A, Supplementary Fig. 1). Meanwhile, Cdc42 KO mice only showed disrupted polarity in the villi, consistent with previous findings[5], while Kras mice exhibited no noticeable differences in the intestine epithelia compared to that of WT mice (Fig. 1A). The Cdc42 KO/Kras mice rapidly lost 15% of their body weight within 96 h after TAM injections, and 100% of them did not survive beyond 7 days. In contrast, Cdc42 KO or Kras mice showed comparable body weight and survival rate to WT mice (Fig. 1B, C). Western blot analyses of isolated intestinal epithelia confirmed the loss of Cdc42 protein in the intestines of Cdc42 KO and Cdc42 KO/Kras mice, and the activation of Kras downstream targets p-ERK and p-AKT (Thr308) in the intestines of Kras and Cdc42 KO/Kras mice were apparent (Fig. 1D). To examine the cause of lethality of Cdc42 KO/Kras mice, we performed a TUNEL assay to analyze cell death in the intestine and found a significantly elevated TUNEL$^+$ cell death in the villi area of the Cdc42 KO/Kras small intestine that were absent in mice of the other genotypes (Fig. 1E, F, Supplementary Fig. 1). Co-staining with a nutrient-absorptive enterocyte marker ALPi suggests that enterocytes are the main cell type in the intestine dying in the Cdc42 KO/Kras mice, which could explain the rapid loss of body weight and subsequent lethality of the mice (Fig. 1E). The expression of the tight junction marker ZO-1 is also altered in Cdc42 KO/Kras mice, indicating disrupted tight junctions (Fig. 1E).

### Cdc42 KO/Kras G12D double mutations cause reduced ISCs, mis-localized Paneth cells and disrupted basal-lateral polarity

To examine the changes of the major cell types in the Cdc42 KO/Kras intestine, we performed immunostaining of multiple cell-type-specific markers in the fixed duodenum samples. The percentage of epithelial cells expressing the enterocyte marker ALPi in Cdc42 KO/Kras mice was reduced by over 60%, while the epithelial cells became round and dissociated, suggesting a loss of epithelial function (Fig. 2A, G). The number as well as the mucin secretion-related genes in goblet cells and hormone expression-related genes in enteroendocrine cells did not significantly change in the Cdc42 KO/Kras mice (Fig. 2B, C, Supplementary Fig. 2A, B). However, the localization of Paneth cells in CDdc42 KO/Kras mice was altered, with Paneth cells found in the villi area versus the crypt area, a pattern seen in the Cdc42 KO mice, whereas all Paneth cells reside at the bottom of the crypts in the WT mice (Fig. 2D, H). Similar to Cdc42 KO, the ISC population in Cdc42 KO/Kras crypts reduced significantly (Fig. 2E, I), and basolateral polarity was disrupted in the Cdc42 KO/Kras villi (Fig. 2F, Supplementary Fig. 2C). mRNA expression of the ISC marker *Lgr5*, the Paneth cell marker *Defa5*, and the proliferation marker *Cyclin D1* was also reduced in the Cdc42 KO/Kras intestine, suggesting an impaired ISC and its niche functions (Fig. 2J).

### Single-cell RNAseq analyses reveal elevated inflammation, necroptosis, and disrupted adhesion junctions in Cdc42 KO/Kras G12D mutant intestine

To explore the underlying mechanism of the disrupted morphology and the loss of enterocytes and ISCs in Cdc42 KO/Kras mice, we isolated villi/crypt cells from WT, Cdc42 KO, Kras, and Cdc42 KO/Kras mice and performed high-throughput single-cell mRNA-sequencing (scRNAseq) and uniform manifold approximation and projection (UMAP) clustering analysis, a nonlinear dimensionality-reduction technique[15]. Unbiased clustering of isolated villi/crypt cells from all four samples, followed by cluster annotation based on cell marker expressions[5] revealed at least seven intestinal cell types: Enteroendocrine cells, enterocytes, Goblet cells, Paneth cells, ISCs, transit-amplifying (TA) cells, and tuft cells, as well as major immune cell types: macrophages, B cells, and T cells (Supplementary Fig. 3A–C). We performed gene set enrichment analysis (GSEA) and found that acute inflammation, necroptosis, and response to wound pathways were significantly elevated in the combined gut cell populations (enterocytes, enteroendocrine cells, Goblet cells, Paneth cells, ISCs, TA cells, Tuft cells) in the Cdc42 KO/Kras mice, whereas cellular respiration pathway signature genes were reduced (Fig. 3A). Similar pathway changes were observed in the three gut cell subtypes, i.e. enterocytes, ISCs/TA cells and Goblet cells, when analyzed individually (Supplementary Fig. 3C). This suggests that the Cdc42 KO/Kras mice are experiencing an acute inflammatory response and necroptosis-mediated cell death. Adhesion junction pathways and actin filament pathways are also altered in the Cdc42 KO/Kras intestine (Fig. 3A). Three major necroptosis pathway marker genes: *tnfrsf23, bok, and tnf* were enriched in the enterocyte cluster in the Cdc42 KO/Kras genotype (Fig. 3B, Supplementary Fig. 3B), which is consistent with the TUNEL staining pattern in the villi. In addition, we found that YAP signaling, a known regulator of ISC regeneration upon injury[16,17] and a critical mediator of the ISC-to-TA cell fate transition[5], is elevated in Cdc42 KO and Cdc42 KO/Kras TA cells/ISCs (Fig. 3C), suggesting a potential role in the observed inflammation and necroptosis/cell death phenotypes.

### Cdc42 KO/Kras G12D mutant shows reduced proliferation and increased necroptosis and inflammation in intestine

To verify the relevance of the scRNAseq analyses, particularly related to the inflammatory response and necroptosis changes in the Cdc42 KO/Kras intestine, we first examined proliferation using Ki67 and

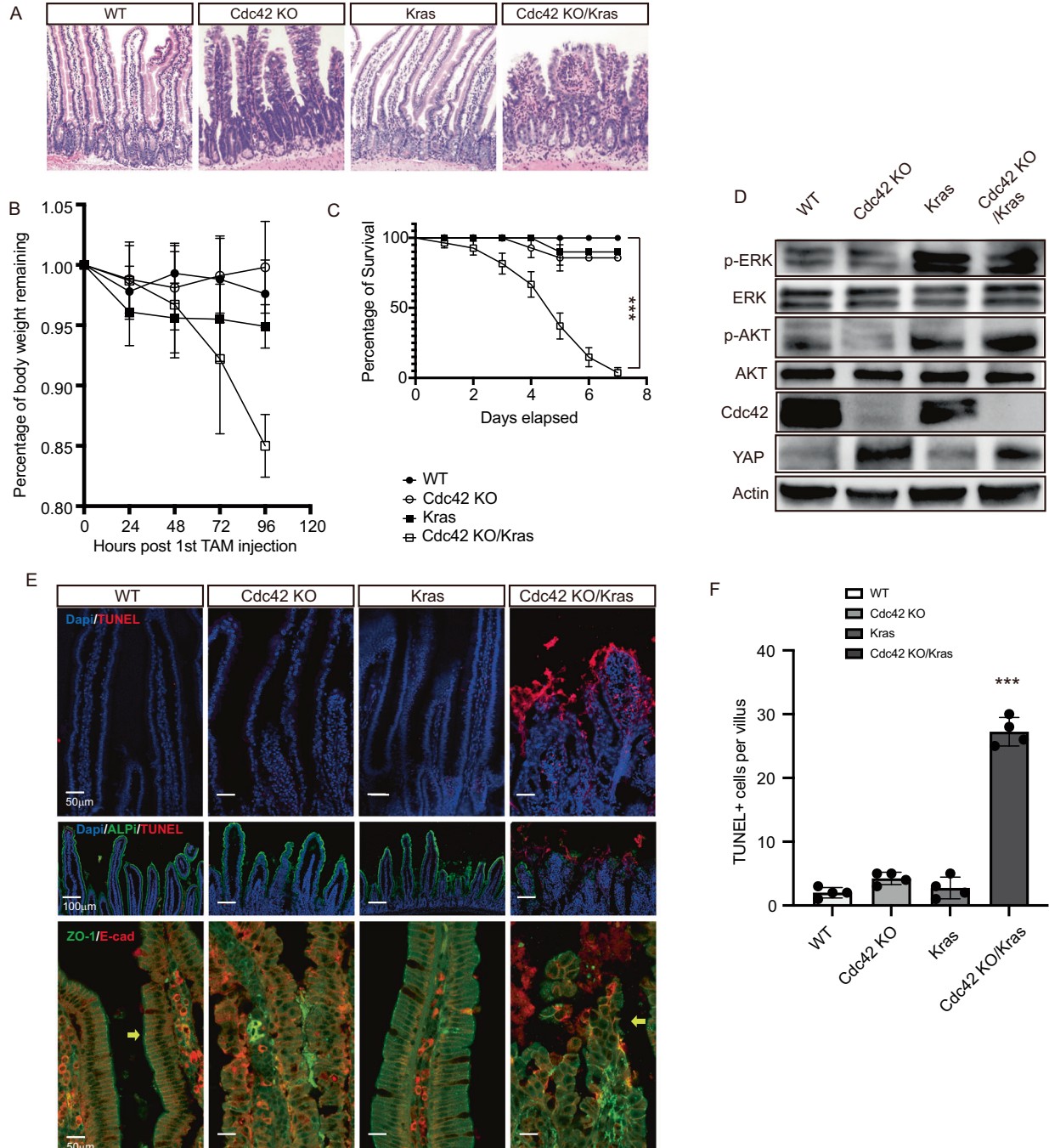

**Fig. 1 | Cdc42 KO combined with oncogenic Kras G12D disrupts epithelial morphology and causes lethality. A** Three to four-month-old mice were injected with TAM once per day for 3 days, and then were sacrificed at 96 h after the 1st TAM injection. Representative images of H&E staining of duodenal sections. Data are representative of at least three independent experiments. **B** Percentage of body weight remaining at 24–120 h after 1st TAM injection. Data are mean ± SD, WT $n = 4$ mice, Cdc42 KO $n = 3$ mice, Kras $n = 3$ mice, Cdc42 KO/Kras $n = 6$ mice. Source data are provided as a Source Data file. **C** Percentage of survival at 1–8 days after 1st TAM injection. Data are mean ± SE. Log-rank (Mantel-Cox) test, ∗∗∗ $P < 0.0001$, WT $n = 13$ mice, Cdc42 KO $n = 17$ mice, Kras $n = 14$ mice, Cdc42 KO/Kras $n = 28$ mice. Source data are provided as a Source Data file. **D** Small intestinal tissues (crypts and villi) were isolated and lysed for Western blotting. F-actin is the loading control. Representative images of 3 independent biological repeats. **E** Representative images of immunofluorescence staining of duodenal sections. Data are representative of at least three independent experiments. Scale bars, 50 μm. Source data are provided as a Source Data file. **F** Quantification of the number of TUNEL+ cells per villi/crypt. Data are mean ± SD; ∗∗∗$p < 0.005$. Source data are provided as a Source Data file.

phospho-histone H3 staining. We found that crypt cell proliferation was significantly elevated in the Cdc42 KO and Kras mice compared to WT, but was tempered in Cdc42 KO/Kras mice (Fig. 4A, B, G). Apoptosis, marked by cleaved caspase-3, was not evident in the Cdc42 KO/Kras mice (Fig. 4C), suggesting that the observed cell death is not due to programmed cell death. Caspase-1 plays a critical role in the inflammatory response by mediating the activation of pro-inflammatory cytokines, particularly interleukin-1β (IL-1β)[18,19]. We found that caspase-1 is enriched in the entire epithelium of the Cdc42 KO/Kras duodenum, but less so in the distal small intestine (ileum) or colon (Fig. 4D, H, Supplementary Fig. 1). Additionally, *interleukin-1β* (*IL-1β*) and *tumor necrosis factor* (*tnf*), two other key pro-

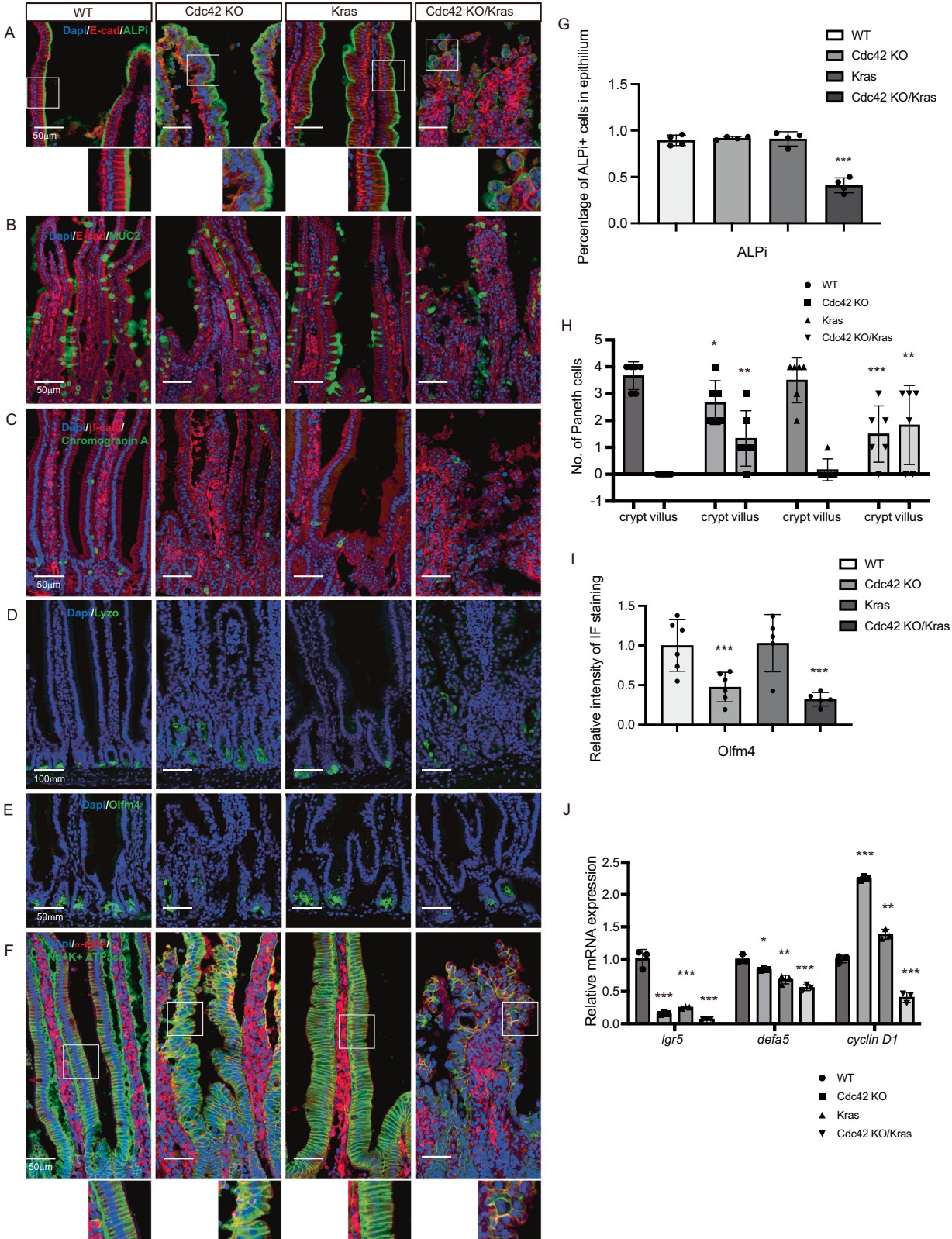

inflammatory cytokines that play critical roles in the inflammatory response, are also elevated in the Cdc42 KO/Kras mice (Fig. 4I). Necroptosis markers pMLKL and RIP[20,21] are overexpressed in the Cdc42 KO/Kras epithelium in small intestine and colon (Fig. 4E, F, H, Supplementary Fig. 1), consistent with the scRNA-seq results. Immunofluorescence staining for macrophages, T cells, and TNFα shows their accumulation primarily in the top region of the villi, including capillaries and lacteals (Supplementary Fig. 4A), while flow cytometry analysis reveals that multiple immune cell populations are increased in the Cdc42 KO/Kras mice, including macrophages, neutrophils, T cells, and B cells (Fig. 4J, Supplementary Fig. 4B). These findings indicate that necroptosis and an elevated immune response, characterized by increased immune cells accumulation and overexpression of inflammatory cytokines and necroptosis markers,

**Fig. 2 | Cdc42 KO/Kras G12D mice have reduced ISCs, mis-localized Paneth cells and disrupted basal-lateral polarity. A–F** Representative images of immuno-fluorescence staining of duodenal sections. Data are representative of at least three independent experiments. Scale bars, 50 μm. **G** Quantification of the percentage of ALPi+ cells in each villus. Data are mean ± SD; two-tailed unpaired Student's *t*-test, ***$p$ = 1.713E-05, $n$ = 4 mice for each group tested. Source data are provided as a Source Data file. **H** Quantification of the number of Paneth cells per villus/crypt. Data are mean ± SD; two-tailed unpaired Student's *t*-test, *$p$ = 0.030, **$p$ = 0.010, ***$p$ = 0.001, **$p$ = 0.012, $n$ = 6, 2 crypts/villus per mouse from 3 mice for each group tested. Source data are provided as a Source Data file. **I** Quantification of the relative intensity of IF staining of Olfm4. Data are mean ± SD; two-tailed unpaired Student's *t*-test, ***$p$ = 0.006, ***$p$ = 0.002, WT $n$ = 6 mice, Cdc42 KO $n$ = 6 mice, Kras $n$ = 5 mice, Cdc42 KO/Kras $n$ = 5 mice. Source data are provided as a Source Data file. **J** Quantification of relative mRNA expression. Data are mean ± SD; two-tailed unpaired Student's *t*-test, ***$p$ = 0.0005, ***$p$ = 0.0008, ***$p$ = 0.0003, *$p$ = 0.014, **$p$ = 0.004, ***$p$ = 4.694E-06, ***$p$ = 0.0005, **$p$ = 0.002, ***$p$ = 0.0004, $n$ = 3 mice for each group tested. Source data are provided as a Source Data file.

contribute to the severe deterioration of the intestine epithelia of the Cdc42 KO/Kras mice.

## Inhibition of necroptosis or IL1/TNFα-mediated inflammation rescues defects of Cdc42 KO/Kras G12D mutant intestine

The defects observed in the Cdc42 KO/Kras mice, i.e., weight reduction, disrupted villous mucosal structure, altered expression and localization of tight junction proteins, inflammatory response, and necrosis, are typical characteristics of the NEC mouse models[22]. To date, the pathogenesis of NEC is not entirely understood, but recent studies indicate that it is multifactorial, involving the immaturity of the intestinal barrier system, ischemic injury to the intestine, and hyperinflammation[13]. To understand whether necroptosis (and injury to the intestine) and/or hyperinflammation is a leading cause of NEC-like defects in the Cdc42 KO/Kras intestine, we performed a "rescue" experiment using the necroptosis inhibitor Nec-1s[23] or the IL-1R inhibitor anakinra[24,25].

Administration of the necroptosis inhibitor Nec-1s to Cdc42 KO/Kras mice improved the general morphology of the duodenum and restored the integrity of the villous mucosal structure (Fig. 5A). pMLKL staining along with reduced TUNEL signals confirmed the efficacy of necroptosis inhibition in the Cdc42 KO/Kras epithelium (Fig. 5B, G, H), suggesting that the inhibition of necroptosis is sufficient to prevent the intestine cell death phenotype in the Cdc42 KO/Kras mice. Consistently, the Cdc42 KO/Kras mice treated with Nec-1s showed reduced body weight loss and exhibited improved survivability (Fig. 5I, Supplementary Fig. 5A). Interestingly, the populations of ISC and proliferative cells in the crypts were also restored by Nec-1s (Fig. 5D), suggesting that necroptosis in the villi impacted the TA cell/ISC populations in the crypts. To further assess the ISC function, we performed an enteroid colony growth assay using Nec-1s−rescued crypts. While Nec-1s treatment restored ISC survival and allowed formation of enteroid spheres compared to untreated Cdc42 KO/Kras ISCs, their clonogenic potential was not fully rescued (Supplementary Fig. 5D), indicating that the rescue of ISC number and survival is incomplete and that additional functional defects persist beyond cell survival. Not surprisingly, the necroptosis inhibitor restored the enterocyte population in the Cdc42 KO/Kras epithelium (Supplementary Fig. 5B), whereas the mis-localization of Paneth cells and loss of basolateral polarity were not rescued (Fig. 5E, Supplementary Fig. 5B). In addition, macrophages and neutrophils were reduced but caspase-1 remained overexpressed by Nec-1s (Fig. 5C, F, H, J, Supplementary Fig. 5C).

To further explore other forms of programmed cell death, such as pyroptosis, we examined cleaved gasdermin D and found that it was upregulated in Cdc42 KO/Kras mice but was not rescued by necroptosis inhibition (Supplementary Fig. 5B), indicating an activation of the pyroptosis pathway in the double mutant mice. This observation is consistent with the finding that although inhibition of necroptosis reduced epithelial cell death and extended survival (Fig. 5I), approximately 60% of Nec-1s−treated mice still exhibited a delayed mortality, suggesting that unresolved pyroptosis may also contribute to lethality in Cdc42 KO/Kras mice.

Similar to Nec-1s, intraperitoneal injection of the IL-1R inhibitor anakinra or anti-TNFα antibody also restored the morphology of Cdc42 KO/Kras mice (Fig. 6A, Supplementary Fig. 6C) and reduced necroptosis, inflammation and cell death in the villi (Fig. 6B, G, H, Supplementary Fig. 6A–C), implicating IL-1/TNFα pathway in the overall phenotypes of the double mutants and suggesting a positive feed-forward loop between IL-1/ TNFα-mediated hyperinflammation and necroptosis. Notably, IL-1R inhibition effectively suppressed immune cell−derived IL-1β and TNFα expression (Supplementary Fig. 6A), indicating that the inflammatory signaling and necroptosis phenotypes are largely driven by inflammation-derived micro-environmental cues. In contrast, epithelial-intrinsic defects, such as disrupted polarity marked by Par3 and Na⁺/K⁺-ATPase disorganization (Fig. 6D, Supplementary Fig. 6A) and altered epithelial morphology (Fig. 6A), were not restored, suggesting that disrupted adhesion junctions and polarity pathways are intrinsic to the epithelium.

Enterocytes, ISCs, and proliferating cells were also restored by the IL-1 inhibitor treatment (Fig. 6C, E, F, I, J), while the localization of Paneth cells and basolateral polarity were not rescued (Fig. 6D, E, Supplementary Fig. 6B). Given the critical niche-supporting role of Paneth cells for ISCs, we considered whether their mislocalization might contribute to ISC depletion. Previous studies have demonstrated that Paneth cells are dispensable for ISC maintenance and function[26]. Consistent with this, Paneth cell mislocalization alone is not sufficient to cause ISC loss in our models, as ISC recovery was observed in the above rescue experiments with Nec-1s or IL1R inhibitor, despite persistent Paneth cell mislocalization.

These results from the administration of necroptosis or IL-1R inhibitors in Cdc42 KO/Kras mice demonstrate a significant restoration of intestinal morphology and proliferation, highlighting the interplay between necroptosis and hyperinflammation in the pathogenesis of the NEC-like defects. Despite improvements in enterocyte and ISC populations, mislocalization of Paneth cells and loss of basolateral polarity remained unaffected, indicating that necroptosis or inflammation is likely downstream of epithelium loss of polarity and intestinal disruption.

## ISC-specific Cdc42 KO/Kras G12D mutations cause similar NEC-like defects as intestinal epithelia-targeted mutations

We observed a loss of stemness marker Olmf4 in the Cdc42 KO/Kras mice, which can be restored by inhibition of necroptosis and inflammation. To investigate the possible effect of Cdc42 KO and Kras activation from ISCs leading to the NEC-like phenotypes, we utilized an inducible ISC-specific Cdc42 KO/Kras line. This was achieved by breeding Cdc42 flox/flox mice with Olfactomedin-4 (Olfm4)-CreER knock-in mice[27–29] to generate Olfm4-CreER; Cdc42 flox/+ (hereafter referred to as O-WT) mice, Olfm4-CreER; Cdc42 flox/flox (hereafter referred to as O-Cdc42 KO) mice, Olfm4-CreER; Kras LSL-G12D + (hereafter referred to as O-Kras) mice, and Olfm4-CreER; Cdc42 flox/flox; Kras LSL-G12D + (hereafter referred to as O-Cdc42 KO/Kras) mice. Following TAM injection, the O-Cdc42 KO/Kras mice experienced weight loss and lethality as observed in the villin-CreER driven Cdc42 KO/Kras mutant strain (Fig. 7I), albeit in a slower time course. We examined the O-Cdc42 KO/Kras mice five days after TAM injection and observed similar defects as those seen in the villin-CreER Cdc42 KO/Kras mice, including disrupted epithelium, increased cell death in the villi, loss of polarity, and loss of

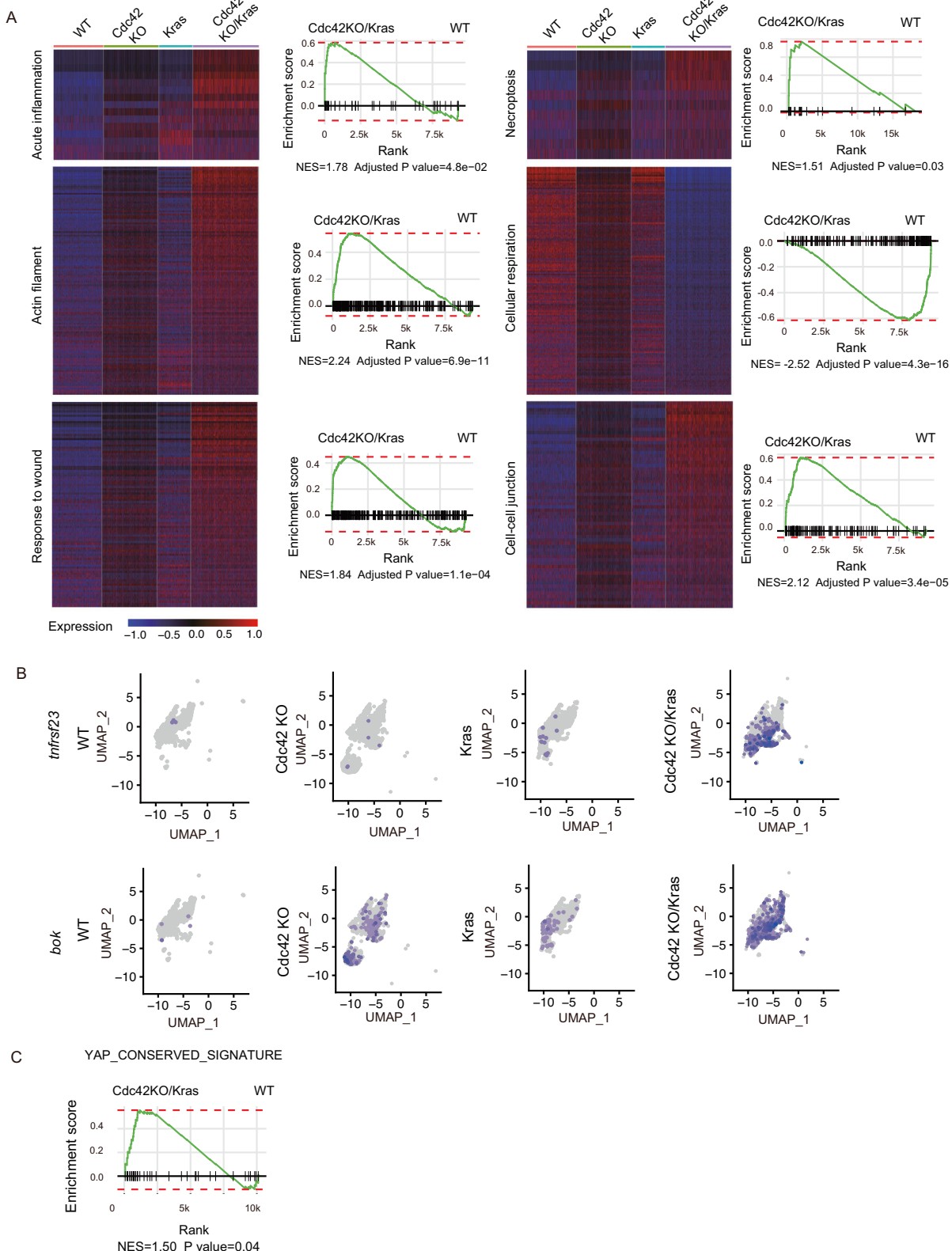

**Fig. 3 | Single-cell RNAseq analysis reveals elevated inflammation, wound response, necroptosis, and disrupted adhesion junction in Cdc42 KO/Kras G12D intestine. A.** Heatmap of curated genes in the acute inflammation, necroptosis, actin filament, cellular respiration, response to wound and cell-cell junction in WT, Cdc42 KO, Kras and Cdc42 KO/Kras all intestinal cell type clusters. Columns indicate individual cells; rows indicate genes. GSEA pathway enrichment maps for corresponding heatmap pathways in all intestinal cell type clusters, for Cdc42 KO/ Kras vs WT. **B** Expression level of necrosis marker *tnfrsf23* and *bok* in all intestinal cell type clusters. **C** GSEA pathway enrichment maps for YAP signaling pathways in TA cells/ISCs, for Cdc42 KO/Kras vs WT.

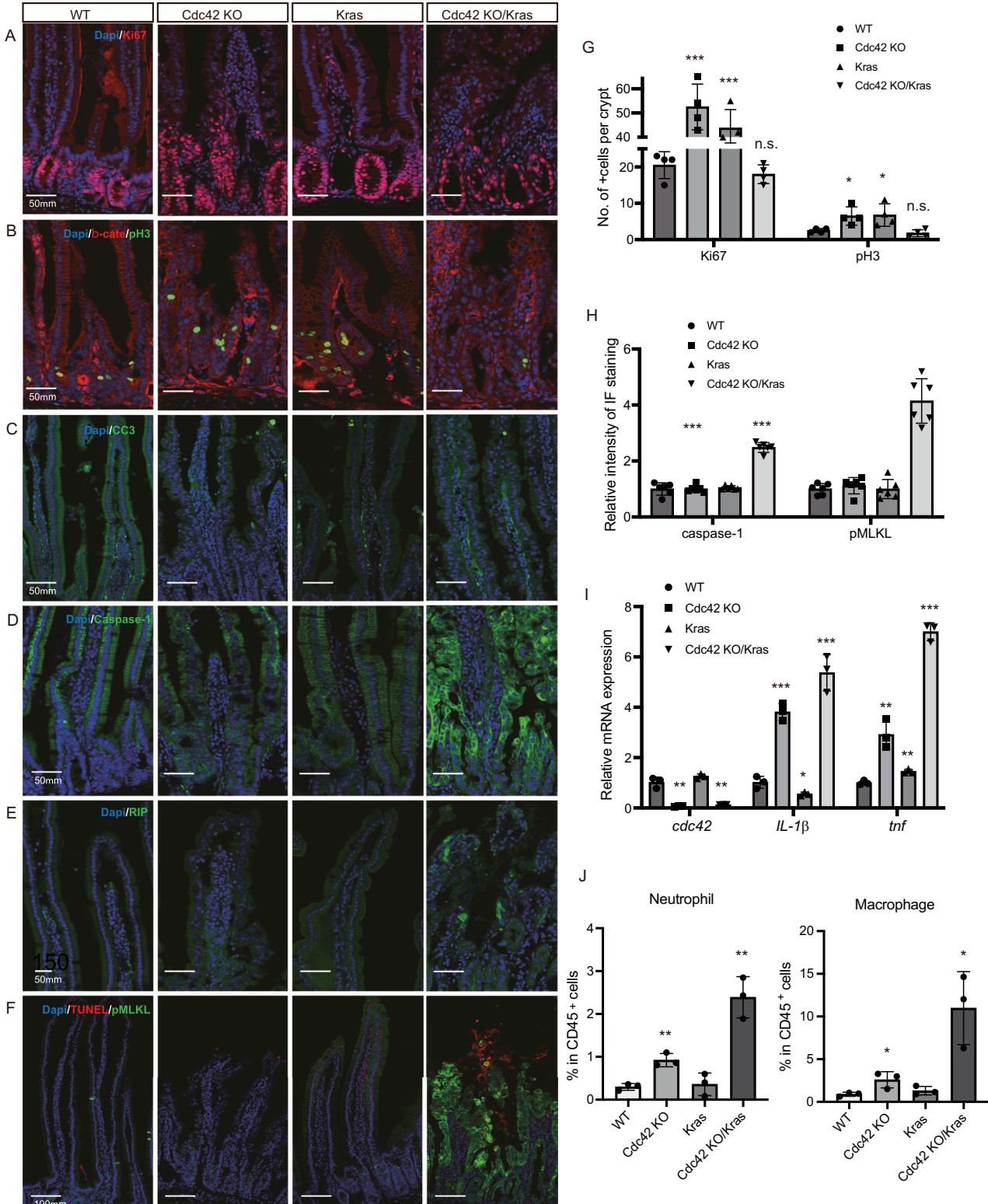

**Fig. 4 | Cdc42 KO/Kras G12D mice have reduced proliferation and increased necroptosis and inflammation. A–F** Representative images of immunofluorescence staining of duodenal sections. Data are representative of at least three independent experiments. Scale bars, 50 μm. **G** Quantification of the number of Ki67 +/pH3+ cells per crypt. Data are mean ± SD; two-tailed unpaired Student's *t*-test, ***i = 0.0007, ***p = 0.0016, n.s. > 0.05, *p = 0.021, *p = 0.036, n.s. > 0.05, *n* = 4 mice for each group tested. Source data are provided as a Source Data file. **H** Quantification of the relative intensity of IF staining of caspase-1 and pMLKL. Data are mean ± SD; two-tailed unpaired Student's *t*-test, ***p = 3.002E-09, ***p = 1.099E-

07, *n* = 6 mice for each group tested. Source data are provided as a Source Data file. **I** Quantification of relative mRNA expression. Data are mean ± SD; two-tailed unpaired Student's *t*-test, **p = 0.0015, **p = 0.0019, ***p = 0.0003, *p = 0.033, ***p = 0.0006, **p = 0.005, **p = 0.003, ***p = 9.442E-06, *n* = 3 mice for each group tested. Source data are provided as a Source Data file. **J** Quantification of the percentages of neutrophil and macrophage among live CD45+ cells in isolated single cells from duodenal villus/crypts. Data are mean ± SD; two-tailed unpaired Student's *t*-test, **p = 0.003, ***p = 0.002, *p = 0.04, *p = 0.02, *n* = 3 for each genotype. Source data are provided as a Source Data file.

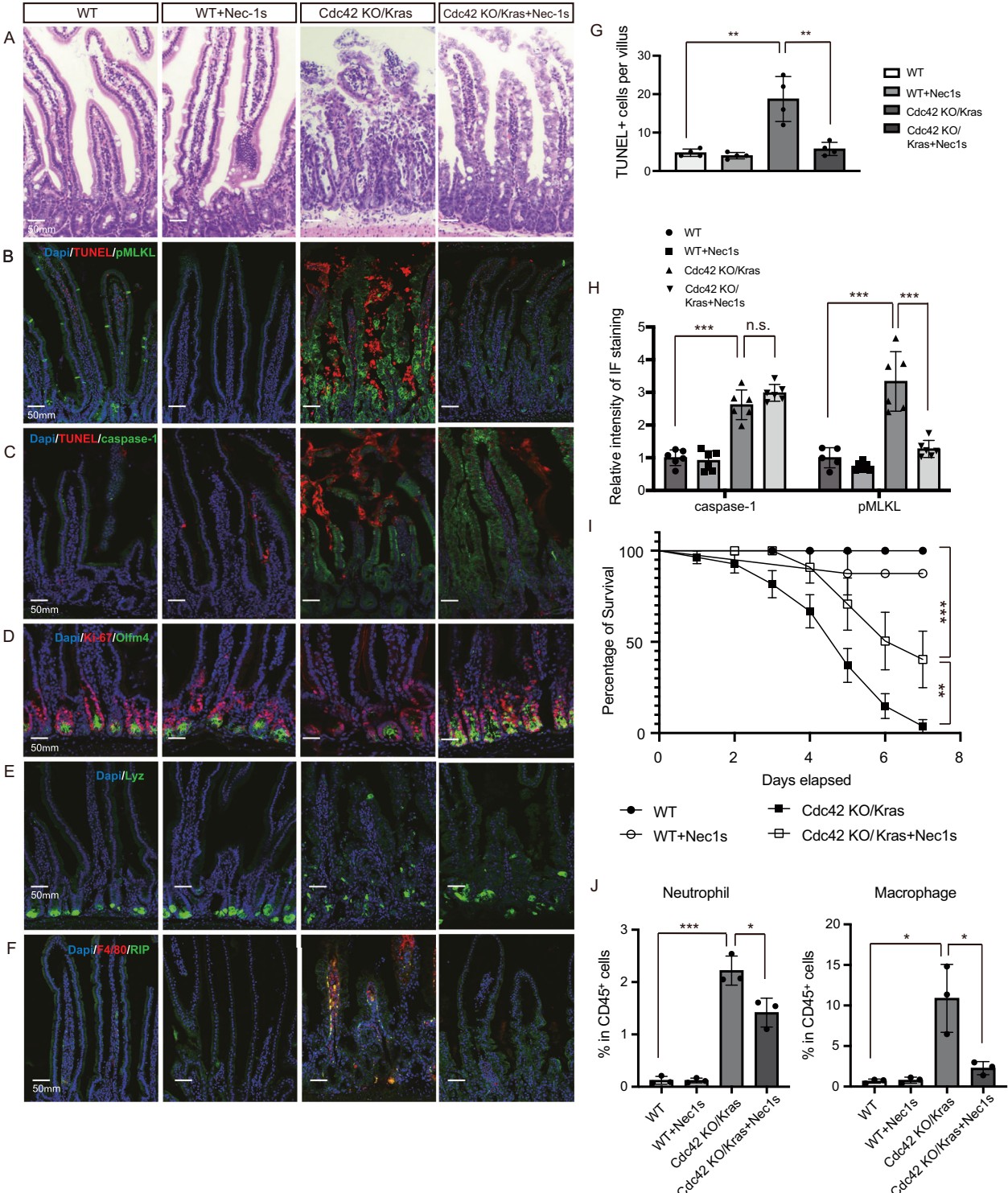

**Fig. 5 | Necroptosis inhibitor Nec-1s rescues Cdc42 KO/Kras G12D intestinal defects. A** Representative images of H&E staining of duodenal sections. Data are representative of at least three independent experiments. Scale bars, 50 μm. **B**–**F** Representative images of immunofluorescence staining of duodenal sections. Data are representative of at least three independent experiments. Scale bars, 50 μm. **G** Quantification of the number of TUNEL+ cells per villi/crypt. Data are mean ± SD; two-tailed unpaired Student's $t$-test, **$p$ = 0.003, **$p$ = 0.005, $n$ = 4 mice for each group tested. Source data are provided as a Source Data file. **H** Quantification of the relative intensity of IF staining of caspase-1 and pMLKL. Data are mean ± SD; two-tailed unpaired Student's $t$-test, ***$p$ = 1.567E-05, n.s. = 0.117,

***$p$ = 0.0004, ***$p$ = 0.0003, $n$ = 6 mice for each group tested (except $n$ = 5 for WT pMLKL). Source data are provided as a Source Data file. **I** Percentage of survival at 1–8 days after 1st TAM injection. Data are mean ± SE. Log-rank (Mantel-Cox) test, ∗∗∗$P$ = 0.0004, ∗∗$p$ = 0.007, WT $n$ = 7 mice, WT+Nec-1s $n$ = 8 mice, Cdc42 KO/Kras $n$ = 28 mice, Cdc42 KO/Kras+Nec-1s $n$ = 13 mice. Source data are provided as a Source Data file. **J** Quantification of the percentages of neutrophil and macrophage among live CD45+ cells in isolated single cells from duodenal villus/crypts. Data are mean ± SD; two-tailed unpaired Student's $t$-test, ***$p$ = 0.0002, *$p$ = 0.023, *$p$ = 0.014, *$p$ = 0.025, $n$ = 3 for each genotype. Source data are provided as a Source Data file.

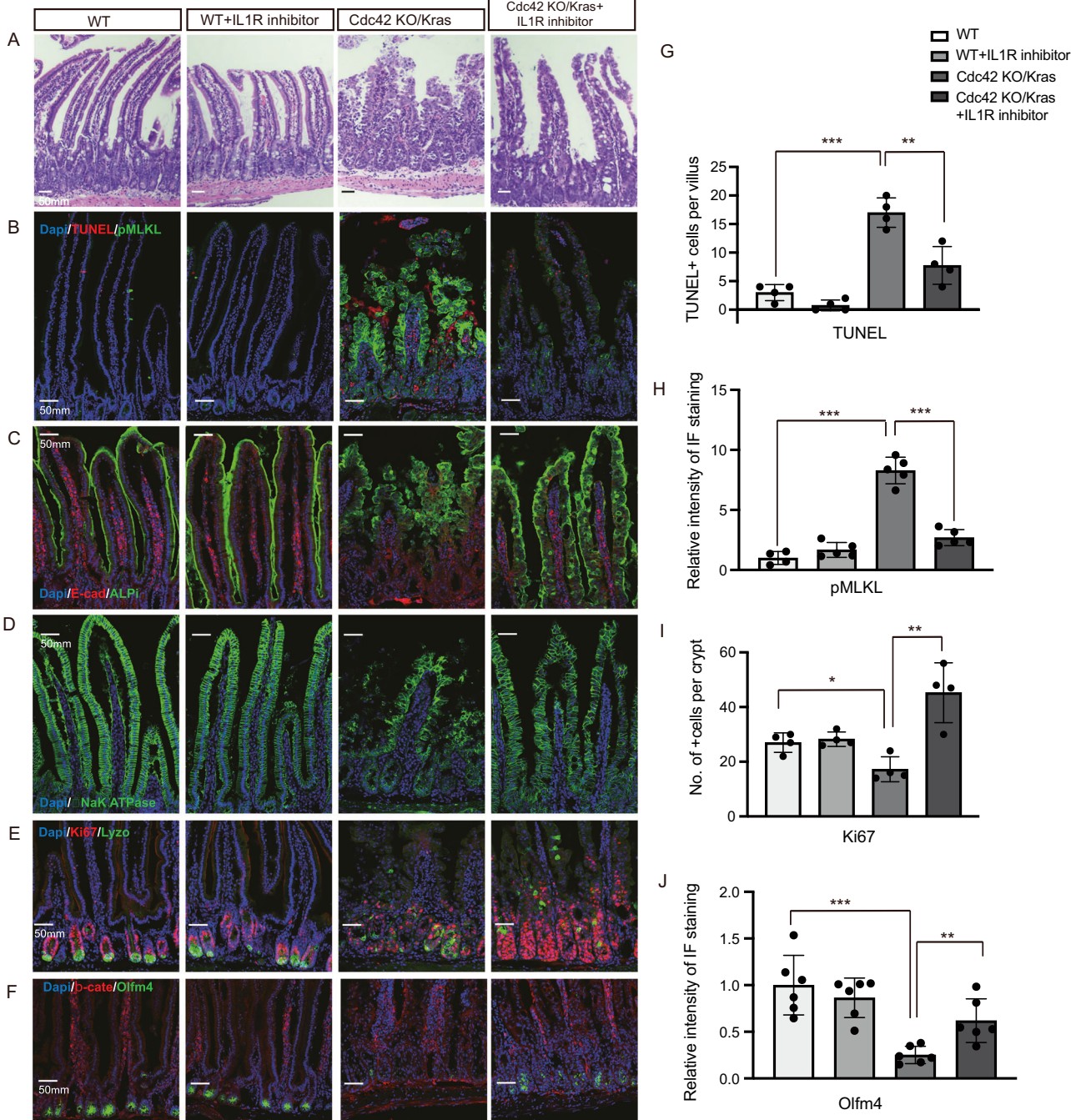

**Fig. 6 | IL-1R inhibition rescues Cdc42 KO/Kras G12D intestinal defects.**
**A** Representative images of H&E staining of duodenal sections. Data are representative of at least three independent experiments. Scale bars, 50 μm.
**B–F** Representative images of immunofluorescence staining of duodenal sections. Data are representative of at least three independent experiments. Scale bars, 50 μm. **G** Quantification of the number of TUNEL+ cells per villi/crypt. Data are mean ± SD; two-tailed unpaired Student's $t$-test, ***$p = 7.702E-05$, **$p = 0.0045$, $n = 4$ mice for each group tested. Source data are provided as a Source Data file.
**H** Quantification of the relative intensity of IF staining of pMLKL. Data are mean ±

SD; two-tailed unpaired Student's $t$-test, ***$p = 6.348E-06$, ***$p = 1.072E-05$, $n = 5$ mice for each group tested (except $n = 4$ for WT pMLKL). Source data are provided as a Source Data file. **I** Quantification of the number of Ki67+ cells per crypt. Data are mean ± SD; two-tailed unpaired Student's $t$-test, *$p = 0.015$, **$p = 0.0032$, $n = 4$ mice for each group tested. Source data are provided as a Source Data file.
**J** Quantification of the relative intensity of IF staining of Olfm4. Data are mean ± SD; two-tailed unpaired Student's $t$-test, ***$p = 0.0003$, **$p = 0.005$, $n = 6$ mice for each group tested. Source data are provided as a Source Data file.

ISC stemness marker (Fig. 7A–D). These results suggest that loss of Cdc42 combined with Kras activation in ISCs is sufficient to initiate NEC-like defects, as the mutant ISCs can differentiate into mature intestinal tissues to propagate defects to the enterocytes in the villi.

Since YAP signaling mostly resides in the crypt ISC/TA cells and is elevated in the Cdc42 KO/Kras intestine (Fig. 3C), we further examined

whether the increased YAP activity plays a role in the development of NEC-like defects. Intraperitoneal administration of the YAP inhibitor Verteporfin to O-Cdc42 KO/Kras mice was able to rescue the general morphology of the intestinal epithelium (Fig. 7E) and reduce cell death, necroptosis, and inflammatory response of the mutant (Fig. 7F, J, K, Supplementary Fig. 7A). YAP inhibition also restored crypt cell

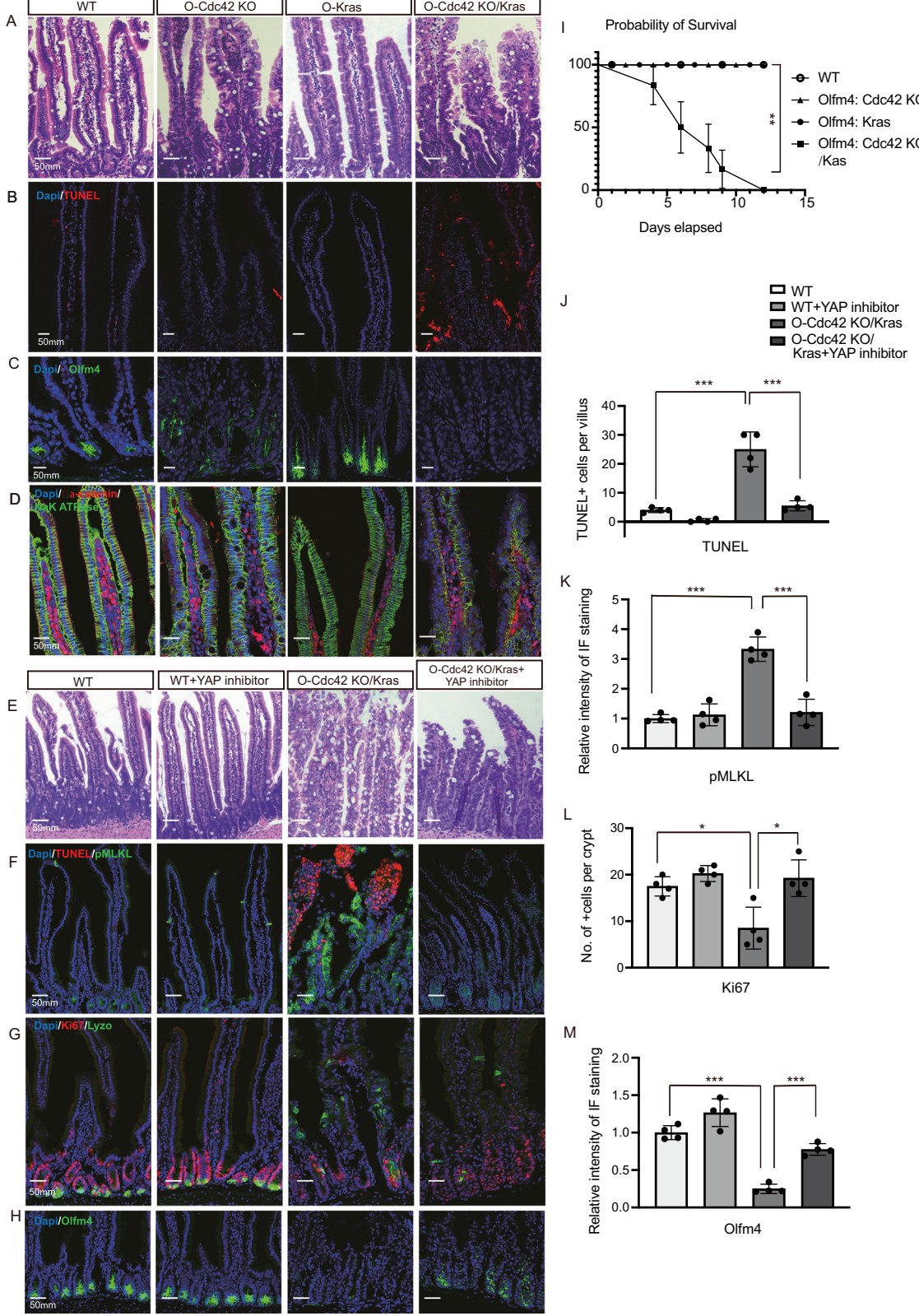

proliferation (Fig. 7G, L) and ISC marker expression (Fig. 7H, M), though it did not correct the mislocalization of Paneth cells (Fig. 7G). Flow cytometry analysis revealed that the macrophages in O-Cdc42 KO/Kras mice were reduced (Supplementary Fig. 7B). The nuclear YAP expression was not impacted by Nec-1s or IL1R inhibitor, suggesting that elevated YAP signaling is upstream of the necroptosis and IL1/TNFα-mediated inflammatory feed forward loop (Supplementary

Fig. 7C). These results indicate that elevated YAP signaling in the ISC/TA crypt cells contributes to the Cdc42 KO/Kras phonotypes in the villi. They provide insights into the mechanism underlying the intestinal tissue damage - elevated YAP signaling by Cdc42 KO, aided by a strong oncogenic cue such as Kras G12D mutation from the crypt progenitor cells, can induce inflammation and necroptosis, leading to NEC-like defects.

**Fig. 7 | ISC-selective Cdc42 deletion/Kras G12D activation causes similar NEC-like defects as intestinal epithelium targeting which can be partially rescued by YAP inhibition. A** Representative images of H&E staining of duodenal sections. Data are representative of at least three independent experiments. Scale bars, 50 µm. **B−D** Representative images of immunofluorescence staining of duodenal sections. Data are representative of at least three independent experiments. Scale bars, 50 µm. **B−F** Representative images of H&E staining of duodenal sections. Data are representative of at least three independent experiments. Scale bars, 50 µm. **F−H** Representative images of immunofluorescence staining of duodenal sections. Data are representative of at least three independent experiments. Scale bars, 50 µm. **I** Percentage of survival at 1–12 days after 1st TAM injection. Data are mean ± SE. Log-rank (Mantel-Cox) test, ∗∗ $P = 0.0037$, WT $n = 4$ mice, Olfm4: Cdc42 KO $n = 6$ mice, Olfm4: Kras $n = 6$ mice, Olfm4: Cdc42 KO/Kras $n = 6$ mice. Source data are provided as a Source Data file. **J** Quantification of the number of TUNEL+ cells per villi/crypt. Data are mean ± SD; two-tailed unpaired Student's t-test, ***$p = 0.0004$, **$p = 0.0008$, $n = 4$ mice for each group tested. Source data are provided as a Source Data file. **K** Quantification of the relative intensity of IF staining of pMLKL. Data are mean ± SD; two-tailed unpaired Student's t-test, ***$p = 3.550E\text{-}05$, ***$p = 0.0004$, $n = 4$ mice for each group tested (except $n = 4$ for WT pMLKL). Source data are provided as a Source Data file. **L** Quantification of the number of Ki67+ cells per crypt. Data are mean ± SD; two-tailed unpaired Student's t-test, *$p = 0.011$, *$p = 0.012$, $n = 4$ mice for each group tested. Source data are provided as a Source Data file. **M** Quantification of the relative intensity of IF staining of Olfm4. Data are mean ± SD; two-tailed unpaired Student's t-test, ***$p = 1.005E\text{-}05$, ***$p = 4.203E\text{-}05$, $n = 4$ mice for each group tested. Source data are provided as a Source Data file.

## Discussion

Cell polarity molecules play a critical role in maintaining the structural and functional organization of intestinal epithelial tissues, and their disruptions are associated with the morphologic phenotypes of epithelial tumors[30–32]. On the other hand, KRAS, a key oncogene, is frequently mutated in intestinal cancer and drives tumor growth and resistance to therapy[7,33]. However, the interplay between loss of polarity and oncogenic KRAS in intestinal cancer remains poorly understood. We hypothesized that loss of function in Cdc42 and related polarity regulators, such as DLG5, SCRIB, and LLGL1, may corroborate with oncogenic KRAS in colon cancer progression. However, an analysis of data from The Cancer Genome Atlas (TCGA) found that the loss-of-function mutations of polarity-associated molecules occur infrequently (-1% to 6%), and they are mutually exclusive with mutations in KRAS in human colon cancer patients (Supplementary Fig. 8A, B). Furthermore, among the rare cases where mutations in both KRAS and these polarity molecules were present, patients demonstrated significantly worse survival rates, raising the possibility that the combined mutations are detrimental to survival, which may not result in epithelial transformation and progression to cancer.

To examine the above possibility, we have investigated the functional consequence and associated mechanism of the double mutation of Cdc42 loss and oncogenic Kras in an inducible mouse model. Surprisingly, mice with Cdc42 depletion combined with Kras activation exhibited rapid weight loss and died within a few days. Further histological analysis revealed profound defects, including reduced body weight, disruption of the villous mucosal structure of the intestines, heightened inflammatory responses, and necrosis, in the Cdc42KO/Kras double mutant mice. These phenotypes mimic that of NEC, a severe gastrointestinal condition that primarily affects premature or low-birth-weight infants characterized by inflammation of the intestinal tissue, which can progress to tissue necrosis and, in severe cases, perforation of the intestinal wall[14,34]. These findings suggest that the loss of Cdc42-regulated epithelial polarity is mutually exclusive for survival with oncogenic Kras mutation; rather, they lead to a severe disruption of intestinal barrier integrity, inflammation, and catastrophic tissue damage, phenotypes of a NEC-like disease. Our study highlights the complex interplay between polarity regulators and oncogenic Kras signaling pathways and suggests that a mutual exclusivity of these mutations may arise as a natural selection mechanism to prevent the deleterious effects in mammals (Fig. 8).

Cdc42 is known to regulate intestinal epithelia integrity and intestinal stem cell (ISC) function by mediating Hippo signaling[5,17,35]. Loss of Cdc42 disrupts small intestine polarity and causes hyperactivation of YAP signaling in TA and ISC cells[4,5]. The elevated YAP drives the transition of ISCs into TA cells and promotes intestinal hyperplasia, a precursor to pathological changes[5]. In the Cdc42KO/Kras double mutant mice, single-cell RNAseq studies saw increased YAP, inflammation, and necroptosis gene signatures in the small intestine cells. The heightened YAP activity appears to play a pivotal role in the NEC-like pathology, as a YAP inhibitor treatment could suppress the NEC-like phenotypes. This suggests a synergistic interaction between Cdc42 loss, causing YAP activation and Kras-driven oncogenic signaling, leading to severe intestinal damage. These findings implicate YAP as a key mediator of the intestinal phenotype observed in our model.

As YAP signaling is highly concentrated at the bottom of the crypts, where the TA and ISC cells reside[5,36], yet necroptosis and cell death predominantly occur in the enterocytes of the villi, we hypothesized that YAP-dependent signaling of the crypt cells may initiate the NEC-like defects in the villi. In the ISC-intrinsic Olfm4-CreER driven Cdc42 KO/Kras mice, loss of Cdc42 and activation of oncogenic Kras originate in the ISCs. As ISCs mature and differentiate into intestinal enterocyte cells, persistent YAP and Kras activations induce an inflammatory response characterized by the accumulation of neutrophils and macrophages. The resulting immune hyperactivation, combined with morphologic disorganization due to the loss of polarity, drives necroptosis and the NEC-like disease.

A hallmark of NEC is intestinal epithelia necroptosis[34,37]. We found that MLKL-regulated necroptosis and IL-1R-regulated inflammation are two major contributors to intestinal cell death by necroptosis in the Cdc42 KO/Kras mice. Importantly, the inhibition of either necroptosis or IL-1/TNFα inflammation rescues necroptosis and improves the survival of the mice. Interestingly, the ISC population and proliferation in the crypts are also rescued by the inhibition of necroptosis, which mainly happens in the villi. A possible reason is that severe loss of mature enterocytes in the villi triggers ISC differentiation that may exhaust the ISC population, whereas a rescue of enterocyte death may reverse ISC exhaustion. Alternatively, the rescued enterocytes in the villi may dedifferentiate to ISCs in the crypts. The detailed mechanism needs further investigation.

The interaction between necroptosis and IL-1-driven inflammation is a tightly interwoven process, where necroptosis amplifies inflammatory responses, and inflammation influences the initiation or outcomes of necroptosis[20,38,39]. Cytokine release from necroptotic cells can further amplify inflammation[20,39,40], while IL-1 signaling through IL-1R activates downstream NF-κB signaling and inflammatory cytokines such as TNF-α, which upregulates the expression of key necroptotic proteins, including RIPK3 and MLKL[41,42]. Our studies suggest that the interplay between necroptosis and inflammation forms a feed-forward loop that drives the progression of NEC-like defects (Fig. 8). Inhibition of either pathway can disrupt this loop and suppress NEC-like defects.

Importantly, analysis of a duodenal sample from a human NEC patient revealed striking similarities of the pathological features to those observed in our Cdc42 KO/Kras mice, including disrupted epithelial polarity, loss of enterocytes, elevated necroptosis and cell death, and heightened inflammatory responses (Supplementary Fig. 9). These parallels suggest that disruption of epithelial polarity and aberrant epithelial cell signaling may represent a common pathogenic mechanism contributing to NEC-like tissue injury. While our findings

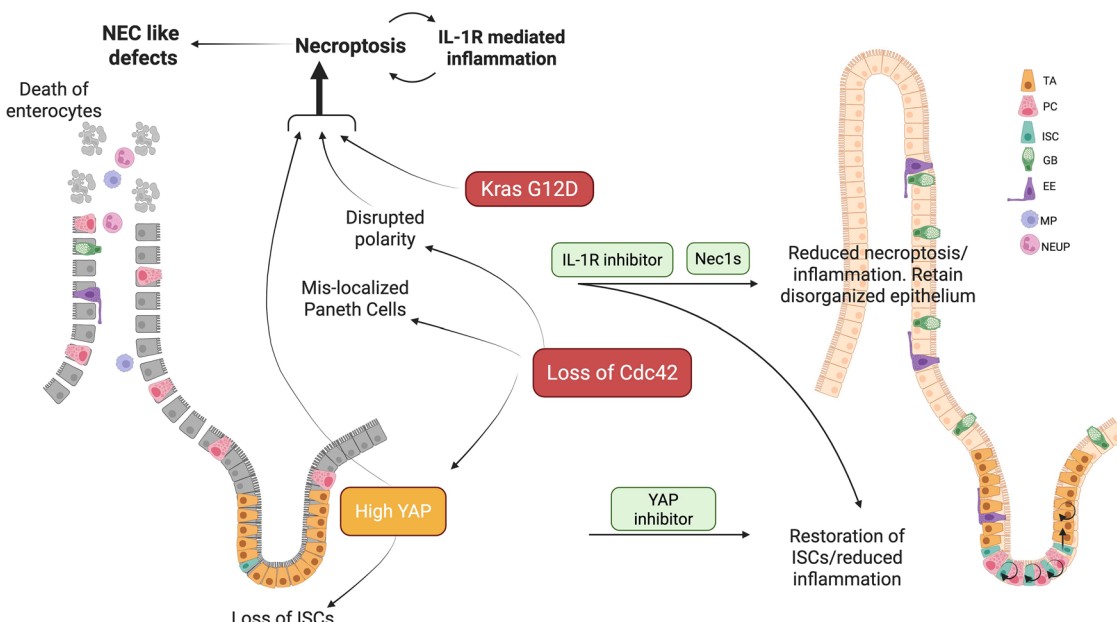

**Fig. 8 | A schematic model of the effects by Cdc42 KO and Kras G12D mutations on mouse intestinal epithelia.** TA: Transit amplifying cells; PC: Paneth cells; ISC: Intestinal stem cells; GB: Goblet cells; EE: Enteroendocrine cells; MP: Macrophage; NEUP: Neutrophil. The graphics are created using BioRender under a publication license.

do not imply that mutations in CDC42 or KRAS occur in human NEC, they raise the intriguing possibility that functional dysregulation of the polarity pathway and stress-activated signaling could exacerbate intestinal barrier breakdown and inflammation. Future studies using neonatal or organoid-based models will be critical to determine whether transient perturbations of epithelial polarity and/or oncogenic signaling may sensitize the intestine to NEC-like injury, thereby providing mechanistic insights for prevention.

Necrotizing enterocolitis (NEC) is a severe gastrointestinal disease that primarily affects preterm infants. In this study, we examined an adult mouse model that unexpectedly showed NEC-like phenotypes, and did not study NEC per se. Due to experimental constraints, we did not explore the mutant combinatorial effects in an infant mouse model, nor in a sequential additive mutation manner. Investigating the interaction in neonatal mice by sequential introduction of polarity and oncogenic Kras mutations could provide valuable insights into the development-related factors and may reveal their potential as biomarkers. Importantly, the NEC-like defects in Cdc42 KO/Kras mice do not imply the presence of similar genetic mutations in human NEC patients, nor do they suggest an involvement of eCdc42 (or other polarity regulators) and/or oncogenic KRAS in human NEC.

## Methods

### Animal studies

All experiments were performed in accordance with guidelines from the Institutional Animal Care and Use Committee (IACUC) at Cincinnati Children's Hospital Medical Center (protocol # IACUC2023-0016). We used a Cdc42$^{flox/flox}$ mouse line based on previous validations[4]. Kras$^{LSL-G12D}$ mice were purchased from Jackson Lab (RRID:IMSR_JAX:008179) and were based on the manufacturer's validations. The Cdc42$^{flox/flox}$ mouse and/or Kras$^{LSL-G12D}$ mice were crossed with Villin-CreERT2 mice provided by Dr. Helen Piwnica-Worms at the University of Texas MD Anderson Center, or Olfm4-IRES-eGFPCreERT2 mice[27] provided by Dr. Hans Clevers at the University of Utrecht. Littermate controls were generated by standard pairings. Animals are housed in an AALAC-accredited barrier facility of CCHMC that meets or exceeds the Animal Welfare Act requirements under a light cycle−on from 6am–8pm, temperature set point at 72 degrees, humidity range of 30%−70%. Euthanasia was performed by CO2 followed by cervical dislocation. To

induce Cre recombinase, 2–3 months old adult mice (mixed gender, no phenotypical difference between male or female mice) were intra-peritoneally injected with 2 mg of tamoxifen (Sigma-Aldrich, Switzerland) dissolved in corn oil for 3 consecutive days from Day 1 to Day 3. Mice were sacrificed and analyzed 5 days after the first injection (Day 5). Nec-1s (5 mg/kg body weight per day), IL-1R inhibitor (anakinra, 50 mg/kg body weight per day), anti-TNFα antibody (2.5 mg/kg body weight, once, on day 0) and YAP inhibitor (Verteporfin, 50 mg/kg body weight per day) were intraperitoneally injected for 5 consecutive days from Day from Day 0 to Day 4.

### H&E staining and tissue preparation

In addition to the above-described mouse tissues, the study used NEC and normal donors' small intestine tissue samples from the Discover Together Biobank at Cincinnati Children's Research Foundation. The intestinal tissues were flushed with PBS and fixed in 10% formalin or 4% paraformaldehyde overnight at 4 degree Celsius. Tissues were then embedded in paraffin or cold optimum cutting temperature embedding medium. Processing of tissues for paraffin-embedding and H&E staining was performed by the CCHMC Digestive Health Center Pathology Core.

### Immunofluorescence and confocal microscopy

Immunofluorescence was performed on frozen sections. Intestinal tissues were flushed with PBS and fixed in 4% paraformaldehyde overnight at 4 °C, cryoprotected in 30% sucrose, embedded in OCT compound, and sectioned at 12 μm. Antibodies for immuno-fluorescence are in Supplementary Data 1 with a 1:250 dilution for the primary antibody and a 1:800 dilution for the secondary antibody. Images were acquired on a Nikon A1R GaAsP Inverted Confocal Microscope, with signal intensity optimized for control samples and the same settings (laser power/gain/offset) used for all tissue for the same experiment.

### Villus/crypt isolation

Sections of intestine (duodenum, ileum, and colon; 5–6 cm in length) were harvested, flushed with ice-cold phosphate-buffered saline (PBS), opened longitudinally, and immersed in fresh ice-cold PBS. The tissue was then cut into five equal pieces and transferred to 10 ml PBS

containing 10 mM EDTA. The tube was shaken vigorously for 20 s, and the supernatant containing debris was discarded before adding fresh PBS–EDTA. The samples were incubated for 30 min at 4 °C, with vigorous shaking for 20 s every 10 min. After incubation, the intestinal pieces were removed, and villus/crypt epithelial cells, together with immune cells, were collected from the PBS–EDTA solution.

## Western blotting

After intestinal epithelium from villi/crypts was isolated, tissues were then homogenized in lysis buffer containing protease and phosphatase inhibitors, sonicated at 4 °C, mixed with 4× sodium dodecyl sulfate loading buffer and heated at 100 °C for 5 min. A Bradford assay was used to determine protein concentration (Bio-Rad). Antibodies used for Western blots are in the Supplementary Data 1 with a 1:400 dilution for the primary antibody and a 1:800 dilution for the secondary antibody. Uncropped and unprocessed scans of the most important blots are in the Source Data file.

## RNA extraction and quantitative real-time PCR

mRNA was isolated from intestinal crypts using an RNA mini kit (Qiagen). RNA concentration was measured by a Nanodrop, and cDNA was produced with a cDNA synthesis kit (ABI). Quantitative PCR was performed using Taqman qPCR master mix (ABI) on a Bio-Rad CFX Opus 96 quantitative reverse-transcription polymerase chain reaction (qRT-PCR) instrument. Relative expression levels were determined by the ΔΔCt method standardized to beta-actin.

## Quantification

phospho-histone H3 (pH3)/Ki67/TUNEL positive cell number was quantified by counting the average number of pH3/Ki67/TUNEL positive cells in one villus and one adjacent crypt (10 villus/crypt per mouse). ALPi+ cell percentage was quantified by counting the number of ALPi-positive epithelial cells in one villus, and then divided by the total number of epithelial cells in the same villus (10 villus per mouse). Relative intensity of Olfm4/Caspase-1/pMLKL was quantified by measuring the YAP immunostaining intensity using ImageJ (Version 2.14.0), which is normalized to Dapi nuclei staining (10 villus/crypt per mouse).

## Statistics

Results are expressed as the mean ± standard deviation. Significance was calculated by Student's t-test. $P < 0.05$ was considered significant. Sample size was ≥ 3 for statistical analysis. They are chosen based on feasible animal cohorts and sufficient statistical power considerations.

## Single-cell suspension preparation

After intestinal epithelium from villi/crypts was isolated, villi/crypts were resuspended in mixed medium comprised of 30% 10x TrypLE and 70% Advanced DMEM/F12, incubated at 37 degree for 30--45 min. Mixing and passaging through a glass pipette was performed every 10 min and suspensions were monitored with a microscope. After disruption of the cell aggregates, cells were filtered through a 40 mm filter and then pelleted by centrifugation (5 min at 300–400 g), washed once with Advanced DMEM/F12, and resuspended in the same medium for 10x genomics scRNA seq.

## Flow cytometry

After single-cell suspensions were prepared, flow cytometry was performed to analyze neutrophils and macrophages based on surface marker expression. Cells were resuspended in staining buffer (PBS containing 2% fetal bovine serum) and incubated with fluorochrome-conjugated antibodies (purchased from BD Biosciences) at 4 °C for 30 min. The following antibodies were used: CD45-FITC (553080), CD11b-APC/Cy7 (557657), Gr-1-BV510 (563040), and F4/80-PE (565410). After staining, cells were washed with staining buffer, and DAPI (Thermo Fisher Scientific, D1306) was used to exclude dead cells.

Samples were acquired on a BD LSRFortessa flow cytometer, and data were analyzed with BD FACSDiva or FlowJo software. Neutrophils were defined as CD45+CD11b+Gr-1+ cells, and macrophages as CD45+F4/80+ cells.

## RNA Library Prep and Sequencing

After single cell suspension, a 3′ Gene Expression Library was prepared for scRNA seq by the CCHMC DNA Genotyping and Sequencing Core, using "Chromium Single Cell 3′ Reagent Kits 3v3.1" by 10x Genomics. The protocol by 10x Genomics in the CG000183 Rev B user guide was strictly followed during the construction of the library, and 14 cycles were run during section 2.2 cDNA amplification. The NovaSeq 6000/NovaSeq X Plus Sequencer was used with the sequencing read parameters designated in the user guide.

## Single-cell data processing

Samples were sequenced on the Novaseq 6000 platform (RRID:SCR_016387). Resulting fastqs were processed through the CellRanger pipeline v5.0.1 (RRID:SCR_017344) using 10X Genomics' mm10-2020-A reference genome, with the setting include introns set to false.

Background reads were removed using decontX from the celda package using the filtered barcodes as the cells to keep and the remaining cells in the raw barcode matrix as the background[43]. Doublets were minimized using DoubletFinder (RRID:SCR_018771)[44]. The R v4.4 (RRID:SCR_001905) library Seurat v4.4 (RRID:SCR_016341)[45] was used for cell type clustering and marker gene identification. Cells expressing >500 genes were retained for downstream analysis. Each sample was normalized SCTransform v2 (RRID:SCR_022146), using the glmGamPoi method and the number of RNA molecules per cell was regressed out. Samples were integrated with common anchor genes using the rPCA method to minimize sample-to-sample variation. Cell clusters were determined by the Louvain algorithm using a resolution of 0.2. UMAP dimension reduction was done using the first thirty principal components. Marker genes for each cell type were calculated using the Wilcoxon Rank Sum test, returning only genes that are present in a minimum of 25% of the analyzed cluster[5]. The similar goblet and Paneth cells were further differentiated by subsetting and reclustering the cell population containing both cell types in an unbiased manner. This resulted in a small group of cells separated from the rest that were enriched with canonical Paneth markers, Lyz1, Defa17, Defa22, Defa24, and Ang4. These cells were annotated as Paneth cells and the rest as goblet cells (Supplementary Fig. 3B). GSEA 4.3.3 (RRID:SCR_003199) was used for pathway enrichment[5,46]. Pathway gene lists were obtained through the Molecular Signatures Database (MSigDB: RRID:SCR_016863). Pathway gene expression heatmaps were plotted using Seurat's DoHeatmap function using values scaled on the mean.

## Reporting summary

Further information on research design is available in the Nature Portfolio Reporting Summary linked to this article.

# Data availability

All data are included in the Supplementary Information or available from the authors, as are unique reagents used in this Article. The raw numbers for charts and graphs are available in the Source Data file whenever possible. Source data are provided with this paper. The scRNA-seq data generated in this study were deposited at the Gene Expression Omnibus (RRID:SCR_005012) under accession number GSE294390 (GSM8903735, GSM8903736, GSM8903737, GSM8903738). The hyperlinks for these data are. GSE294390(study). GSM8903735(WT). GSM8903736(CDC42 KO). GSM8903737(Kras). GSM8903738(CDC42 KO/Kras Source data are provided with this paper.

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

## Acknowledgments

We thank the Cincinnati Children's Hospital Medical Center Discover Together Biobank for support of this study, as well as participants and their families, whose help and participation made this work possible. We thank the expert technical support of James F. Johnson. This work was partly supported by NIH P30 DK078392 (L.D.), NIH U54 DK126108 (Y.Z.), NIH R01 AG063967 (Y.Z.), and NIH R01 CA278756 (Y.Z.).

## Author contributions

Z.Z. conceived the project, designed experiments, performed experiments, carried out data analyses and interpretation, and wrote the paper with input from the rest of the authors. C.F. performed the flow cytometry study. R.J. assisted in performing experiments and manuscript preparation. P.S. provided human NEC samples for analysis. M.A. performed bioinformatic analysis on the single-cell RNAseq data. Y.Z. provided overall supervision of the study, conceived the project, designed experiments, and wrote the paper.

## Competing interests

The authors declare no competing interest.
