## [Transparent Peer Review file · Nature Communications]

Loss of polarity by Cdc42 depletion and oncogenic Kras activation in the mouse intestinal epithelia lead to a necrotizing enterocolitis (NEC)-like disease

Corresponding Author: Dr Yi Zheng

Version 0:

Reviewer comments:

Reviewer #1

(Remarks to the Author)

Cdc42 is a key regulator of epithelial polarity and stem cell maintenance, while oncogenic Kras drives proliferation and disrupts epithelial homeostasis. Although human data suggest these mutations are mutually exclusive to colon cancer, Zheng Zhang and colleagues use inducible Villin-CreER and Olfm4-CreER mouse models to show that combined Cdc42 deletion and Kras activation cause severe intestinal defects resembling necrotizing enterocolitis (NEC), including villous disruption, inflammation, loss of intestinal stem cells (ISCs), and enterocyte necrosis. The authors highlight the synergistic effects of Cdc42 loss and oncogenic Kras activation in the intestinal epithelium. Further single-cell RNA-seq reveals altered polarity programs, hyperactive IL-1 signaling, and necroptosis, which are partially reversible by targeting YAP, IL-1R, or necroptosis pathways. Overall, this study highlights a pathogenic mechanism where disrupted polarity cooperates with oncogenic signaling to impair intestinal homeostasis, providing insight into NEC-like pathology and epithelial vulnerability under dual stress conditions. However, the authors need to address following issues before the manuscript can be considered for publication.

1. Is the lethality more closely related to damage in specific regions (e.g., small intestine vs. colon)? Although the authors mention that both are affected in Fig. 1, are there any regional differences?
2. Fig. 2B-C shows that the numbers of goblet cells and enteroendocrine cells remain unchanged. However, given the disrupted epithelial architecture and loss of polarity, could their functions also be impaired? Have the authors considered performing functional analyses of these cells, such as assessing mucus secretion or hormone expression?
3. In Fig 2F, the authors report disrupted polarity in the villus region; however, a systematic analysis of key polarity proteins (e.g., Par3, aPKC, E-cadherin) is lacking. A more comprehensive assessment of polarity markers is recommended to strengthen the evidence supporting the loss of epithelial polarity, a central phenotype in this study.
4. Additionally, could the mislocalization of Paneth cells be a primary contributor to the observed reduction in intestinal stem cells (ISCs)? The authors are encouraged to explore the causal relationship between these two phenomena, potentially by assessing Paneth cell function, such as lysozyme secretion, to determine whether their niche-supporting role is compromised.
5. In Fig S3A, the cluster annotations need clarification, including the marker genes used to distinguish each cell subtype. For example, how were Paneth and goblet cells differentiated at the single-cell level? The macrophage cluster in the UMAP plot also appears unusual. Was CD68 or any other specific macrophage marker highly expressed in that cluster?
6. In Fig 3A, the GSEA analysis was conducted on all intestinal epithelial clusters combined. Are there subtype-specific differences in pathway enrichment across different intestinal cell types?
7. While the scRNA-seq data point to activated inflammatory signaling, necroptosis, and disrupted adhesion junction pathways, it remains unclear whether these changes are cell-autonomous or driven by immune cell-derived microenvironmental cues. Is there any evidence to distinguish epithelial-intrinsic changes from immune-mediated effects?
8. Given the strong enrichment of *tnf*, *tnfrsf23*, and *bok* in the enterocyte cluster, have the authors tested whether these pathways directly cause cell death? For instance, could inhibition of TNF signaling (e.g., anti-TNF antibodies) or genetic knockout strategies rescue the enterocyte death phenotype?
9. The authors argue that the absence of cleaved caspase-3 suggests non-apoptotic cell death, but have they considered other forms of programmed cell death, such as pyroptosis? Since caspase-1 can also trigger pyroptosis beyond its role in IL-1 β maturation, this warrants further exploration.
10. The authors report a general increase in immune cell populations. Have they examined the spatial distribution of these

immune cells within the intestinal tissue? Do specific immune cell types (e.g., T cells vs. macrophages) localize to distinct pathological regions, which could help better understand the inflammatory response?

11. Can the authors further explore whether YAP signaling plays a role in regulating inflammation or immune responses in this model? While YAP is well known for its role in stem cell biology, it has also been implicated in immune modulation. An integrated discussion of YAP signaling with inflammation and necroptosis pathways could be valuable.

12. Although the number of enterocytes and ISCs appears restored upon treatment, it remains unclear whether their functions are fully recovered. Have the authors considered assessing the functional status of these cells, such as nutrient absorption, stemness gene expression profiles, or clonogenic potential, to provide deeper biological relevance to the observed rescue?

Reviewer #3

(Remarks to the Author)

The authors demonstrate that simultaneous deletion of Cdc42 and activation of oncogenic Kras (G12D) in intestinal epithelial cells or intestinal stem cells induces NEC-like pathology in mice, challenging traditional views of oncogenic cooperation in intestinal transformation. This finding is novel. However, the study leans heavily on phenotypic observations rather than mechanistic insights.

While Cdc42 regulates epithelial polarity and Kras mutations drive intestinal tumorigenesis, the dual genetic manipulation in epithelial or stem cells to model NEC-like pathology questions its relevance to human disease. Human NEC, primarily affecting premature infants, is linked to underdeveloped barrier function, microbial dysbiosis, immature immunity, and feeding practices, not somatic genetic mutations. Although the authors reference TCGA data showing mutual exclusivity between KRAS mutations and polarity gene defects in colon cancer, it is unclear if similar alterations occur in NEC patients. To improve clinical relevance, the authors should analyze human NEC tissue samples for KRAS mutations or expression changes in polarity regulators like CDC42, DLG5, or Scrib.

The single-cell transcriptomic analysis lacks statistical robustness. In Figure 3, GSEA plots omit adjusted P-values, and necroptosis signaling enrichment in the Cdc42 KO/Kras group relies on just three genes with a normalized enrichment score (NES) below 1.5. This is insufficient to confirm pathway activation. The authors should reanalyze the data with proper statistical corrections (e.g., FDR-adjusted P-values) and ensure NES exceeds 1.5 to meet standard thresholds for pathway enrichment.

Treatment with necroptosis inhibitor (Nec-1s), IL-1R antagonist (Anakinra), and YAP inhibitor (Verteporfin) reduces NEC-like phenotypes in Cdc42 KO/Kras mice, but the relationships between IL-1 signaling, necroptosis, and YAP activation remain ambiguous. Are these independent responses to polarity loss and oncogenic stress, or part of a sequential cascade? Additional experiments clarifying the upstream–downstream dynamics of these pathways would strengthen the manuscript.

The study uses both epithelial cell–specific and intestinal stem cell–specific double mutant models, both exhibiting NEC-like pathology. However, it fails to clarify whether these defects stem from aberrant intestinal stem cell behavior or differentiated epithelial cells. Which model better reflects human NEC? The phenotype-driven nature of the study would benefit from deeper investigation into the cellular origin of these defects to enhance mechanistic understanding.

Version 1:

Reviewer comments:

Reviewer #1

(Remarks to the Author)

The authors have addressed most of the technical issues. Through supplementary experiments and in-depth analysis, they have substantially strengthened the study's mechanistic insights. The reviewer has no additional comments.

Reviewer #3

(Remarks to the Author)

No further comment.

Responses to the reviewers' critiques:

Reviewer #1

We thank the reviewer for appreciating that “this study highlights a pathogenic mechanism where disrupted polarity cooperates with oncogenic signaling to impair intestinal homeostasis, providing insight into NEC-like pathology and epithelial vulnerability under dual stress conditions”. The insightful critiques helped us improve the manuscript significantly.

1. “Is the lethality more closely related to damage in specific regions (e.g., small intestine vs. colon)? Although the authors mention that both are affected in Fig. 1, are there any regional differences?”

Answer: We provide the H&E and IF staining of additional regions of small intestine and colon in **Supplementary Figure 1**, which demonstrates that similar morphological defects to that found in duodenum are also present in the distal small intestine (ileum) and the colon of the Cdc42 KO/Kras mice. The results also show that while loss of polarity and necroptosis occur in both small intestine and colon of the double mutant mice, the most severe cell death/damages occur in the small intestine.

2. “Fig. 2B-C shows that the numbers of goblet cells and enteroendocrine cells remain unchanged. However, given the disrupted epithelial architecture and loss of polarity, could their functions also be impaired? Have the authors considered performing functional analyses of these cells, such as assessing mucus secretion or hormone expression?”

Answer: To consider potential effect on goblet cells and enteroendocrine cells, we have compared mucin secretion related genes in goblet cells and hormone expression related genes in enteroendocrine cells by pathway enrichment analyses of the single-cell RNA-seq data. The analysis did not detect significant overall changes in these pathways between Cdc42 KO/Kras cells and WT cells (**Supplementary Figure 2B**). We would like to carry out mucus and hormone secretion tests in future studies.

3. “In Fig 2F, the authors report disrupted polarity in the villus region; however, a systematic analysis of key polarity proteins (e.g., Par3, aPKC, E-cadherin) is lacking. A more comprehensive assessment of polarity markers is recommended to strengthen the evidence supporting the loss of epithelial polarity, a central phenotype in this study.”

Answer: We now include additional polarity marker analyses: E-cadherin staining is shown in **Figure 2B**, Par3, aPKC, and Scribble staining are presented in **Supplementary Figure 2C**. These results further confirm that polarity is disrupted in the intestinal epithelium of Cdc42 KO/Kras mice.

4. “Additionally, could the mislocalization of Paneth cells be a primary contributor to the observed reduction in intestinal stem cells (ISCs)? The authors are encouraged to explore the causal relationship between these two phenomena, potentially by assessing Paneth cell function, such as lysozyme secretion, to determine whether their niche-supporting role is compromised.”

Answer: We agree that Paneth cells provide an important niche support for ISCs. However, multiple studies have demonstrated that Paneth cells are dispensable for ISC survival and function (reviewed in Michaela Quintero and Linda C. Samuelson, Cellular and Molecular Gastroenterology and Hepatology, 2025). Consistent with this, mislocalization of Paneth cells alone is not sufficient to cause ISC depletion in our mouse models, since in our rescue experiments with Nec-1s, IL1R inhibitor, or YAP inhibitor, ISC recovery was observed despite of persistent Paneth cell mislocalization. We now discuss our findings that Paneth cell mislocalization is possibly not the primary cause of ISC loss in our model, in the context of the above cited review article.

5. “In Fig S3A, the cluster annotations need clarification, including the marker genes used to distinguish each cell subtype. For example, how were Paneth and goblet cells differentiated at the single-cell level? The macrophage cluster in the UMAP plot also appears unusual. Was CD68 or any other specific macrophage marker highly expressed in that cluster?”

Answer: We have included a complete list of the marker genes used to define each cell type in the UMAP (provided as a TXT file in Supplemental Materials). In addition, the major marker genes for each cell type are shown in a dot plot (Supplementary Figure 3C). The criteria used to distinguish Paneth cells from goblet cells is provided in Supplementary Figure 3B: The similar goblet and Paneth cells were further differentiated by subsetting and reclustering the cell population containing both cell types in an unbiased manner. This resulted in a small group of cells separated from the rest that were enriched with canonical Paneth markers, *Lyz1*, *Defa17*, *Defa22*, *Defa24*, and *Ang4*. These cells were annotated as Paneth cells and the rest as goblet cells. The main UMAP was reannotated to reflect these cell identities (Supplementary Figure 3A).

For the macrophage cluster, we have observed high expression of macrophage-specific markers, including CD68, CD14, and IL1B, confirming the cluster identity (Supplementary Figure 3C).

6. “In Fig 3A, the GSEA analysis was conducted on all intestinal epithelial clusters combined. Are there subtype-specific differences in pathway enrichment across different intestinal cell types?”

Answer: We have performed various pathway enrichment analyses including acute inflammation, actin filament, response to wound, necroptosis, cellular respiration, and cell–cell junction, in three intestinal cell subtypes: enterocytes, ISCs/TA cells, and Paneth cells, as shown in Supplementary Figure 3. Most pathway changes are consistent across the different cell subtypes, with a few exceptions - for example, cellular respiration in ISCs/TA cells does not appear to change.

7. “While the scRNA-seq data point to activated inflammatory signaling, necroptosis, and disrupted adhesion junction pathways, it remains unclear whether these changes are cell-autonomous or driven by immune cell-derived microenvironmental cues. Is there any evidence to distinguish epithelial-intrinsic changes from immune-mediated effects?”

Answer: To address this question, we have examined IL1 β and TNF α expression in IL1R inhibitory antibody “rescued” samples, which showed that immune cell-derived microenvironmental cues were effectively suppressed by IL1R inhibition in Cdc42 KO/Kras mice (Supplementary Figure 6A). As shown in Figure 6, necroptosis was rescued by anti-IL1R treatment that suppresses inflammation. In contrast, the epithelial-intrinsic defects such as disrupted polarity marked by Par3 and

Na⁺K⁺ATPase expression/organization (Supplementary Figure 6A, Figure 6D) and altered epithelial morphology (Figure 6A), were not restored. These results suggest that inflammatory signaling and necroptosis are largely driven by inflammation-derived microenvironmental cues, whereas disrupted adhesion junction and polarity pathways are epithelial-intrinsic. We now make clear of this point in the text.

8. “Given the strong enrichment of *tnf*, *tnfrsf23*, and *bok* in the enterocyte cluster, have the authors tested whether these pathways directly cause cell death? For instance, could inhibition of TNF signaling (e.g., anti-TNF antibodies) or genetic knockout strategies rescue the enterocyte death phenotype?”

Answer: To address this question, we treated *Cdc42* KO/*Kras* mice with an anti-TNF α antibody and found that inhibition of TNF α effectively rescued enterocyte necroptosis, cell death, proliferation defects, and ISC abnormalities (Supplementary Figure 6C). These results indicate that the strongly enriched TNF α signaling signature in the enterocyte cluster directly contributes to the observed enterocyte death phenotype.

9. “The authors argue that the absence of cleaved caspase-3 suggests non-apoptotic cell death, but have they considered other forms of programmed cell death, such as pyroptosis? Since caspase-1 can also trigger pyroptosis beyond its role in IL-1 β maturation, this warrants further exploration.”

Answer: We have examined cleaved gasdermin D in the mutants and found that it is upregulated in *Cdc42* KO/*Kras* mice but is not rescued by inhibition of necroptosis (Supplementary Figure 5B), indicating an activated pyroptosis pathway. This is consistent with our observation that although inhibition of necroptosis alone reduced cell death and extended the survival of *Cdc42* KO/*Kras* mice (Figure 5), approximately 60% of the mice treated with Nec-1s still had delayed death (Figure 5I), suggesting that unresolved pyroptosis may also contribute to lethality in *Cdc42* KO/*Kras* mice. We now include a discussion of this possibility.

10. “The authors report a general increase in immune cell populations. Have they examined the spatial distribution of these immune cells within the intestinal tissue? Do specific immune cell types (e.g., T cells vs. macrophages) localize to distinct pathological regions, which could help better understand the inflammatory response?”

Answer: We have performed T cell and macrophage marker staining as well as TNF α staining to examine the spatial distribution of cytokine and these immune cells within the intestinal tissue (Supplementary Figure 4B). In *Cdc42* KO/*Kras* mice, the increased TNF α signal, T cells, and macrophages were primarily localized in the top region of the villi, including capillaries and lacteals, suggesting that the immune cells are enriched broadly in the epithelial villi.

11. “Can the authors further explore whether YAP signaling plays a role in regulating inflammation or immune responses in this model? While YAP is well known for its role in stem cell biology, it has also been implicated in immune modulation. An integrated discussion of YAP signaling with inflammation and necroptosis pathways could be valuable.”

Answer: To this end, we have examined the inflammatory response in YAP inhibitor rescued samples and found that YAP inhibition effectively attenuated immune activation in the Cdc42 KO/Kras small intestine (**Supplementary Figure 7A**), suggesting that YAP signaling contributes to the regulation of inflammation in this context. On the other hand, IL1R inhibitor or Nec-1s has no impact on the elevated nuclear YAP expression in Cdc42 KO/Kras mice crypts (**Supplementary Figure 7C, marked by arrow**), suggesting that YAP signaling is upstream of the necroptosis/inflammation feed-forward loop. We now discuss this in the context of the working model (**Figure 8**).

12. “Although the number of enterocytes and ISCs appears restored upon treatment, it remains unclear whether their functions are fully recovered. Have the authors considered assessing the functional status of these cells, such as nutrient absorption, stemness gene expression profiles, or clonogenic potential, to provide deeper biological relevance to the observed rescue?”

Answer: Thanks for the excellent suggestion. While we intend to further expand our studies to the functional characterization of various cell populations beyond that by scRNAseq in future studies, we have performed an enteroid colony growth assay of the Nec-1s rescued crypt samples. We found that while Nec-1s treatment restored the survival of Cdc42 KO/Kras ISCs and allowed them to form enteroid spheres compared to untreated Cdc42 KO/Kras ISCs, their clonogenic potential was not fully rescued (**Supplementary Figure 5D**). This observation suggests that the “rescue” of the crypt cell number/survival is incomplete, and additional defects to the ISCs caused by the genetic mutations are beyond those of cell number/survival function. We now discuss this in the text.

Reviewer #3

I thank the reviewer for appreciating that our study “demonstrates that simultaneous deletion of Cdc42 and activation of oncogenic Kras in intestinal epithelial cells or intestinal stem cells induces NEC-like pathology in mice, challenging traditional views of oncogenic cooperation in intestinal transformation. This finding is novel.” We have performed additional experimentations and clarified key issues about the mechanistic insights and human NEC phenotype comparisons based on the reviewer’s very helpful suggestions.

1. “The study leans heavily on phenotypic observations rather than mechanistic insights.”

Answer: Our study of two combined oncogenic cues, i.e. loss of polarity by Cdc42 KO and oncogenic KRas expression, in mouse models led to an unexpected observation that mimics many of the human NEC-like phenotypes. With additional experimentations performed to address both reviewers’ suggestions, the study has established a mechanistic association of the polarity loss/oncogenic Kras cues with almost all the major effects found in human NEC. We demonstrate by phenotypic “rescues” using inhibitors of IL1R, TNF α , necroptosis, or YAP signaling that the combined oncogenic cues elicit a inflammation response leading to necroptosis of the intestinal enterocytes, which further feed forward to the inflammatory signals to exacerbate the NEC-like phenotypes. Mechanistically, The IL1, TNF α , necroptosis, as well as YAP signaling, are causal to this feed-forward loop. We now clearly depict a mechanistic summary of this model in **Figure 8**

“Graphic Model”. We believe that these pathway-specific causal characterizations provide new mechanistic insights into the observed phenotypes.

2. “While Cdc42 regulates epithelial polarity and Kras mutations drive intestinal tumorigenesis, the dual genetic manipulation in epithelial or stem cells to model NEC-like pathology questions its relevance to human disease. Human NEC, primarily affecting premature infants, is linked to underdeveloped barrier function, microbial dysbiosis, immature immunity, and feeding practices, not somatic genetic mutations. Although the authors reference TCGA data showing mutual exclusivity between KRAS mutations and polarity gene defects in colon cancer, it is unclear if similar alterations occur in NEC patients. To improve clinical relevance, the authors should analyze human NEC tissue samples for KRAS mutations or expression changes in polarity regulators like CDC42, DLG5, or Scrib.”

Answer: We apologize for this confusion. We agree that somatic genetic mutations are not the underlying cause for human NEC. Our work does not implicate somatic genetic mutations as a cause for NEC, nor is it focused on studying human NEC. Rather, we report an unexpected finding that the combined oncogenic cues, i.e. loss of polarity and Kras mutations, lead to NEC-like phenotypes, suggesting a mutual exclusivity between KRAS mutations and polarity gene defects found in colon cancer.

To improve the relevance of our work to human NEC, we have analyzed human intestine tissues from normal and NEC patients by examining the polarity markers (Na⁺/K⁺-ATPase and Scribb), the enterocyte marker ALPi, and the adhesion marker E-cadherin. We observed similar polarity and adhesion defects in the NEC tissues as seen in our Cdc42 KO/Kras mouse model. Enterocytes, the predominant cell type in the NEC small intestine, were also severely impaired, closely recapitulating the epithelial defects observed in the mouse model, supporting that our mouse model is “NEC-like”. We now add a new figure describing all the human NEC phenotypes and changes in **Supplementary Figure 9**. In addition, we have revised the text in the Discussion and the Limitation of the Work that the current studies do not mean to implicate somatic mutations, such as a loss of polarity gene and oncogenic Kras, in human NEC; rather, the study is about the combined genetic mutations causing NEC-like effects.

3. “The single-cell transcriptomic analysis lacks statistical robustness. In Figure 3, GSEA plots omit adjusted P-values, and necroptosis signaling enrichment in the Cdc42 KO/Kras group relies on just three genes with a normalized enrichment score (NES) below 1.5. This is insufficient to confirm pathway activation. The authors should reanalyze the data with proper statistical corrections (e.g., FDR-adjusted P-values) and ensure NES exceeds 1.5 to meet standard thresholds for pathway enrichment.”

Answer: We have expanded the necroptosis gene set for the pathway enrichment analysis, which resulted in an updated NES of 1.51 with an adjusted P value of 0.03. Additionally, we have replaced the P values in **Figure 3** with FDR-adjusted P values to improve statistical robustness.

4. “Treatment with necroptosis inhibitor (Nec-1s), IL-1R antagonist (Anakinra), and YAP inhibitor (Verteporfin) reduces NEC-like phenotypes in Cdc42 KO/Kras mice, but the relationships between

IL-1 signaling, necroptosis, and YAP activation remain ambiguous. Are these independent responses to polarity loss and oncogenic stress, or part of a sequential cascade? Additional experiments clarifying the upstream–downstream dynamics of these pathways would strengthen the manuscript.”

Answer: This question is related to reviewer #1, question #11. We have examined the inflammatory response in YAP inhibitor rescued samples and found that YAP inhibition attenuates immune activation in the *Cdc42* KO/*Kras* intestine (Supplementary Figure 7A). An examination of nuclear YAP expression (Supplementary Figure 7C, marked by arrow) in *Nec-1s* rescued samples and in IL1R antibody rescued samples shows loss of *Cdc42*/polarity-elicited YAP activation is causal for the subsequent hyperinflammation and necroptosis effects, and suppression of IL1-mediated inflammation and necroptosis are circularly associated such that inflammation is upstream of necroptosis, which can feed forward to further exacerbate inflammation. We now discuss this in the context of the working model (Figure 8).

5. “The study uses both epithelial cell–specific and intestinal stem cell–specific double mutant models, both exhibiting NEC-like pathology. However, it fails to clarify whether these defects stem from aberrant intestinal stem cell behavior or differentiated epithelial cells. Which model better reflects human NEC? The phenotype-driven nature of the study would benefit from deeper investigation into the cellular origin of these defects to enhance mechanistic understanding.”

Answer: Our studies by using the intestinal stem cell *Olfm4*-CreER driver to delete *Cdc42* gene and activate oncogenic *Kras* showed similar defects in inflammation, loss of polarity/junctional defects, enterocyte necroptosis, etc, as by using the pan-intestinal epithelial *Villin*-CreER driver. To better compare with human NEC phenotypes, we further analyzed human NEC samples and found that the crypt regions are relatively intact, showing normal proliferation and TA cell populations. The main defects were observed in the villus regions, characterized by increased immune activation, necroptosis, and cell death (Supplementary Figure 9). Therefore, both models of the *Cdc42* KO/*Kras* mice reflect similar epithelial cell and inflammatory defects seen in human NEC, suggesting that while the intestinal epithelia necroptosis and inflammation are the manifested phenotypes, original defects in ISCs can be propagated through differentiation to mimic that of the NEC-like effect in the epithelium. We now discuss this point in Discussion.